# Understanding the Limits of Text-Only Molecular Reasoning: A Case Study in Synthetic Chain-of-Thought Supervision

**Deepa M Korani**
dpkn@novonordisk.com

**Mohammad Madani**
iqom@novonordisk.com

**Josefa Lia Stoisser**
ofsr@novonordisk.com

**Marc Boubnovski Martell**
mbvk@novonordisk.com

**Lawrence Phillips**
lwph@novonordisk.com

**Kristine Deibler**
ktdb@novonordisk.com

## Abstract

While large language models show promise for scientific reasoning, their applicability to molecular property prediction remains unclear. We present Mol2Synth, a controlled study that examines whether synthetic chain-of-thought supervision can allow text-only LLMs to match conventional topological fingerprint methods for prediction of toxicity. Our results reveal fundamental limitations: even with tool-grounded reasoning and optimized representations, our best configuration (F1=0.88) underperforms classical ECFP fingerprints (F1=0.96), suggesting an inherent information bottleneck in textual molecular representations. Through systematic ablations across molecular representations (SMILES vs. IUPAC), data scaling, and tool-grounded generation, we demonstrate that reasoning-augmented fine-tuning stabilizes training and improves performance over zero-shot LLMs and label-only supervision, but cannot overcome structural parsing failures inherent to text-only inputs. Our qualitative analysis reveals that the primary failure mode is not faulty chemical reasoning but unreliable SMILES-to-structure interpretation; a bottleneck that tool integration partially addresses but cannot eliminate. These findings establish both the utility and fundamental limits of synthetic chain-of-thought supervision for molecular tasks, motivating hybrid architectures that combine natural language reasoning with explicit structural encoders.

## 1 Introduction

Large language models (LLMs) have demonstrated strong reasoning capabilities across mathematical, scientific, and domain-specific tasks, and are increasingly trained using supervised fine-tuning (SFT) and chain-of-thought (CoT) supervision to elicit explicit natural-language reasoning (1; 2; 3; 4). More recently, "reasoning" models trained with reinforcement learning using verifiable rewards (RLVR), including GRPO-style approaches, have shown that explicit reasoning traces can further improve performance, albeit at the cost of substantial computational overhead and careful reward design (5; 6; 7).

In scientific domains, and chemistry in particular, CoT-style supervision remains comparatively underexplored. Most prior work emphasizes representation learning or domain-specific pretraining (e.g., MolFormer (8), ChemBERTa (9)) and expensive curation of instructional datasets (e.g., ChemDFM (10)). Chemistry nevertheless offers a uniquely informative testbed for reasoning-based alignment: molecules admit textual representations such as SMILES and IUPAC, expose computable properties, and follow well-defined domain rules (e.g., toxicophores) that enable automated plausibility checks of generated reasoning traces (11; 12; 13).

Motivated by the cost and complexity of RLVR (14; 15), we investigate whether LLM-generated synthetic chemical reasoning traces, filtered by a critic LLM and optionally grounded using cheminformatics tools, can serve as a lightweight alternative to reinforcement learning for chain-of-thought supervised fine-tuning in molecular property prediction (16; 17). We introduce **Mol2Synth**, a text-

only post-training framework designed to support controlled analysis of synthetic reasoning supervision across representations and data regimes. Using **Mol2Synth** we evaluate performance on mutagenicity and toxicity benchmarks from MoleculeNet (18), comparing SMILES, IUPAC, and combined representations, and assessing the impact of grounding generation with cheminformatics tools (19).

As illustrated in Figure 1, **Mol2Synth** consists of four stages: (i) subsampling training pairs using SMILES, IUPAC, or both; (ii) prompting a frontier LLM to generate step-by-step chemical reasoning traces, optionally guided by cheminformatics-derived functional groups and descriptors; (iii) ranking and filtering generated traces using a critic LLM to retain chemically coherent rationales (6); and (iv) fine-tuning a target model with either label-only or reasoning-augmented supervision. Additional methodological and experimental details are provided in Appendix A.

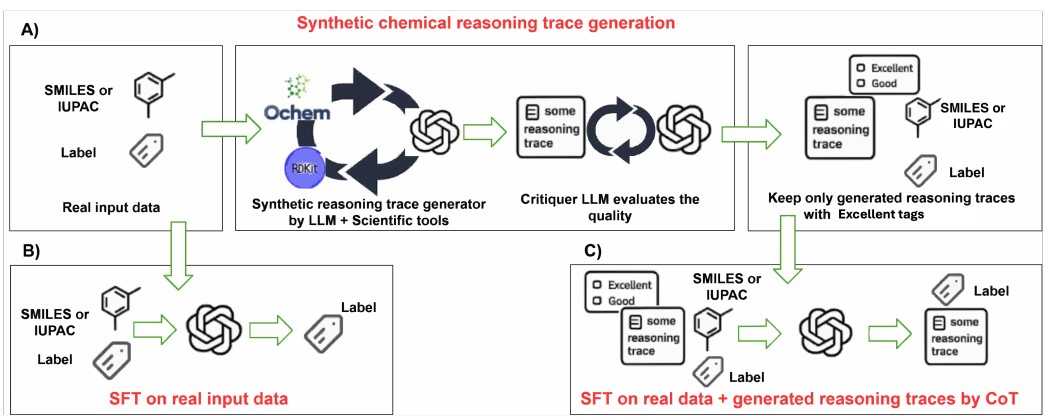

Figure 1: Overview of the **Mol2Synth** workflow: (a) synthetic reasoning trace generation, (b) baseline training using input representations only, and (c) supervised fine-tuning with synthetic chain-of-thought reasoning traces.

This paper makes both positive and negative contributions. On the positive side, we show that synthetic chain-of-thought supervision stabilizes fine-tuning and enables smaller models to outperform larger domain-specialized LLMs including ChemDFM-13B and ether-0. More critically, we demonstrate that even optimized text-only approaches fall short of topological fingerprint methods, revealing representational limits that motivate future hybrid architectures.

Our contributions are as follows:

- **Mol2Synth: Synthetic reasoning as a lightweight alternative to RL.** We introduce **Mol2Synth**, a text-only pipeline for generating, filtering, and integrating synthetic chemical reasoning traces. Fine-tuning with Mol2Synth raises F1 from 0.48 (label-only SFT) to 0.81 on toxicity and surpasses domain-specialized LLMs including ChemDFM-13B and ether-0 at half to one-third the parameter count. Performance saturates at approximately 40% of synthetic data, indicating an inherent information ceiling for text-based supervision.

- **Empirical limits of text-only molecular representations.** IUPAC inputs consistently outperform SMILES (F1 0.88 vs. 0.74 on toxicity), and qualitative analysis with expert annotations traces this gap to structural hallucinations during SMILES parsing rather than deficiencies in chemical reasoning. However, even the best text-only configuration falls short of ECFP fingerprints (F1 0.96), revealing a representational bottleneck that neither scaling nor improved prompting can fully overcome.

- **Tool grounding improves reasoning fidelity but does not close the gap.** Grounding trace generation with cheminformatics tools yields consistent gains across both tasks and models (+7–9% F1 on toxicity) by eliminating error-prone SMILES parsing, yet the performance ceiling relative to explicit classical topological fingerprints persists, confirming that text-based reasoning remains an incomplete proxy for physicochemical signal.

## 2 RELATED WORK

### 2.1 CLASSICAL METHODS FOR MOLECULAR PROPERTY PREDICTION

Classical approaches to molecular property prediction rely on manually engineered molecular fingerprints that convert chemical substructures into fixed-length vector representations. Popular methods include Extended Connectivity Fingerprints (ECFP) (20) and MACCS keys (21), which encode substructure presence or circular neighborhoods within a molecule. These representations are typically used with traditional machine learning models such as Random Forests, Support Vector Machines, or logistic regression to predict properties like toxicity, mutagenicity, or solubility. While effective for many tasks, these approaches are limited by their reliance on pre-defined substructures and lack the ability to capture complex, higher-order chemical interactions.

### 2.2 UNSUPERVISED LEARNING FOR MOLECULAR PROPERTY PREDICTION

Unsupervised Learning overcome some limitations of classical fingerprints by learning representations directly from molecular graphs or textual encodings such as SMILES. Transformer-based architectures, including MolFormer (8), ChemBERTa (9), ChemBERTa-2 (22), and GROVER (23), leverage large-scale pretraining to capture rich chemical semantics. These models have demonstrated improved performance on diverse molecular property prediction benchmarks, particularly under low-data settings, by transferring knowledge from large unlabeled chemical corpora. Graph-based and sequence-based pretraining allows these models to capture subtle structural and functional relationships that classical fingerprints may miss.

### 2.3 LARGE LANGUAGE MODELS FOR MOLECULAR PROPERTY PREDICTION

Recent work has explored the use of large language models (LLMs) to reason about molecules in natural language, either by generating textual explanations or by producing step-by-step chemical reasoning traces. Approaches such as DeepSeek (24), MolReasoner (11), and other reasoning-augmented frameworks demonstrate that LLMs can improve prediction via chain-of-thought supervision, synthetic reasoning traces, or tool-grounded chemical validation. Domain-specialized LLMs have further advanced this direction: ChemDFM (25), a 13B-parameter model pretrained on 34B tokens of chemical literature, demonstrates strong generalist chemistry capabilities that surpass GPT-4 on many chemical tasks, while ether-0 (26), a 24B-parameter reasoning model trained with reinforcement learning on over 640K experimentally-grounded chemistry problems, outperforms frontier LLMs and human experts on molecular design tasks. However, these models come at significant computational cost and rely on large, carefully curated domain-specific datasets; ChemDFM requires 34B tokens of chemical text and 2.7M fine-tuning instructions, while ether-0 demands multiple rounds of reinforcement learning and rejection sampling across hundreds of thousands of verified problems; limiting their accessibility and reproducibility for many research groups. These methods extend beyond purely statistical learning, aiming to incorporate domain knowledge and interpretable reasoning into molecular predictions. However, current studies show that LLM-based reasoning is often limited by the expressivity of textual representations (SMILES or IUPAC names) (27), motivating frameworks like Mol2Synth that integrate synthetic reasoning traces with supervised fine-tuning to improve performance without reinforcement learning.

## 3 METHODS

### 3.1 MOL2SYNTH: SFT ON SYNTHETIC EXPLANATIONS (MOL2SYNTH)

We study supervised fine-tuning on synthetic natural-language explanations (Mol2Synth, referred to as *Mol2Synth*), in which molecular inputs are augmented with LLM-generated reasoning traces prior to training. Unlike label-only SFT, which relies solely on SMILES or IUPAC representations, Mol2Synth introduces intermediate chemical explanations to provide additional contextual supervision about molecular structure, functional groups, and mechanistic hypotheses.

Within Mol2Synth, we consider two instantiations that differ only in how the synthetic explanations are generated: (i) a generalist, text-only generation setting, and (ii) a tool-grounded setting in which explanations are explicitly anchored to cheminformatics tools. In both cases, the downstream student model is trained using identical SFT objectives; the distinction lies exclusively in the source and grounding of the synthetic supervision.

**Mol2Synth (Text-only Synthetic CoT).** In the base Mol2Synth setting, synthetic explanations are generated by a frontier LLM (Gemini 2.5 Pro), prompted to produce step-by-step chemical reasoning based on established chemistry principles, including structure–activity relationships and known toxicity or mutagenicity heuristics. A separate critic LLM (Gemini 2.5 Pro) evaluates each generated explanation and assigns a qualitative score. Only explanations rated *Excellent* are retained for supervised fine-tuning.

**Mol2Synth+tool(Tool-Guided Mol2Synth).** We extend Mol2Synth with a tool-guided variant in which the generator LLM is augmented with subject-matter expert tools. In this setting, the model is instructed to explicitly invoke cheminformatics tools during explanation generation, grounding its reasoning in symbolic chemical analysis rather than free-form text alone. Details of the tools used are provided in the A.4. As in the text-only setting, all explanations are filtered by the same critic LLM, and only those rated *Excellent* are retained.

**Critic-Based Rationale Filtering.** For both Mol2Synth variants, the critic evaluates explanations using five task-specific criteria tailored to toxicity and mutagenicity. Explanations are rated *Good* if at least four out of five criteria are satisfied, and *Excellent* if all five criteria are satisfied. Only *Excellent* explanations are used for training. The full set of evaluation criteria is summarized in Table 1.

Across both generator and critic LLMs, we use a temperature of 0.3. All experiments are repeated across three random seeds (0.1, 0.2, 0.3) to assess robustness; we report the mean and standard deviation across runs.

Table 1: Rationales used to evaluate the LLM on Toxicity and Mutagenicity

| ID | Toxicity rationale | Mutagenicity rationale |
|----|--------------------|------------------------|
| R1 | Is the reasoning chemically sound? | Is the reasoning chemically sound? |
| R2 | Does it identify relevant structure features and toxicophores. | Does it identify relevant structure features known to cause mutagenicity |
| R3 | Does it consider appropriate toxicity mechanisms? | Does it recognize DNA-reactive groups? |
| R4 | Is the logic well and clear connected? | Does it consider metabolic activation pathways? |
| R5 | Are the chemical concepts accurate? | Is the logic clear and well connected |

## 4 EXPERIMENTAL SETUP

**Datasets and Representations.** We evaluate performance on Mutagenicity ($N \approx 6,000$, Ames test) and Toxicity ($N = 7,000$, ClinTox and Tox21) (18; 28). ClinTox provides binary toxicity labels directly. For Tox21, a molecule is labeled toxic if positive in at least 50% of assays, and non-toxic otherwise. To isolate representational bottlenecks, we process all molecules into both SMILES (textual geometry) and IUPAC (semantic nomenclature) formats. Data splits are summarized in Table 5.

**Training Configuration.** We fine-tune DeepSeek-R1 (Distilled Llama 3.1 8B) using LoRA ($r = 16$) with AdamW optimizer (learning rate $1 \times 10^{-4}$) for 50 epochs. We observe consistent trends when replicating experiments with DeepSeek-R1 (Distilled Qwen-7B). Detailed hyperparameters are listed in Table 6, and compute resources are highlighted in Appendix A.2

**Baselines.** We evaluate Mol2Synth and its tool-augmented variant (Mol2Synth+tool) against label-only supervised fine-tuning (SFT-Direct) and a strong structure-aware baseline based on ECFP fingerprints, which serves as a reference for non-textual molecular representations. Additional comparisons to zero-shot LLMs and task-specific molecular models are reported in Appendix A.2. We also conduct expert (medicinal chemist) assessments to evaluate the specificity and usefulness of generated reasoning traces (Appendix A.7)

## 5 RESULTS

### 5.1 QUANTITATIVE RESULTS

As detailed in Table 2, Mol2Synth successfully rescues the DeepSeek-R1 (Distilled LLaMA-3.1-8B) model from the instability of label-only fine-tuning (SFT-Direct), raising F1 scores from a volatile 0.48 to a reliable 0.74. We observe analogous trends on the mutagenicity task (Table 3). Importantly, across both tasks, the tool-grounded variant (Mol2Synth+tool) consistently outperforms the zero-shot baseline of the same underlying model: on toxicity, the LLaMA-8B model improves from a zero-shot F1 of 0.71 to 0.81 after fine-tuning, while the Qwen-7B model improves from 0.67 to 0.72. A similar pattern holds on mutagenicity, where the LLaMA-8B model rises from 0.75 to 0.80. This confirms that synthetic chain-of-thought supervision provides a genuine learning signal beyond what the pretrained model can achieve through prompting alone.

Notably, our fine-tuned models also surpass both ChemDFM-13B and ether-0, domain-specialized chemistry LLMs with 13B and 24B parameters respectively; two to three times the size of our base models. On toxicity, Mol2Synth+tool with LLaMA-8B achieves an F1 of 0.81 compared to ChemDFM's 0.70 and ether-0's 0.71; on mutagenicity, the margins are even wider (0.80 vs. 0.61 and 0.69). This suggests that targeted fine-tuning with structured reasoning traces can be more effective than broad domain pretraining or reinforcement learning for specific property prediction tasks, even with a substantially smaller model.

However, a clear performance ceiling emerges whose severity is task-dependent. On toxicity, even our best configuration—tool-grounded reasoning with IUPAC inputs (Table 4, F1 0.88)—falls short of the ECFP fingerprint baseline (F1 0.96), a gap that persists across all molecular representations and supervision strategies. This suggests that holistic toxicity prediction relies on global physicochemical signals that textual encodings, even when augmented with synthetic reasoning, cannot fully recover.

On mutagenicity, the gap narrows substantially. Mol2Synth+tool with LLaMA-8B (F1 0.80) surpasses ChemBERTa-2 (F1 0.71) and MolFormer (F1 0.63), and achieves F1 competitive with ECFP (F1 0.78), though ECFP retains a higher overall accuracy (0.79 vs. 0.69). This indicates that for tasks where toxicophore recognition and localized substructure reasoning are central, synthetic chain-of-thought supervision can close much of the gap to classical structure-aware models. Taken together, these results suggest that the information bottleneck of text-based molecular reasoning is not uniform: it is most pronounced for properties requiring distributed physicochemical features, and least restrictive when the prediction task aligns with the kind of discrete, rule-based reasoning that chain-of-thought traces naturally express.

### 5.2 ABLATIONS

**1. Impact of Data Scale (Saturation).** To determine whether the observed performance ceiling arises from insufficient supervision, we analyze performance as a function of synthetic data scale (Figure 2). For reasoning-augmented SFT, IUPAC representations enable rapid learning, with performance saturating beyond approximately 40% of the synthetic data fraction. In contrast, label-only fine-tuning (SFT-Direct) becomes increasingly unstable as data scale increases, particularly for SMILES inputs.

We further find that reasoning fidelity is critical: relaxing the critic filter from *Excellent* to *Unfiltered* reduces F1 from 0.73 to 0.64 (Table 7). This confirms that while high-quality synthetic supervision is necessary for stable learning, it remains insufficient to close the gap to structure-aware fingerprint baselines.

**2. Impact of Tool Grounding.** Incorporating cheminformatics tools during trace generation (Mol2Synth+tool) consistently improves performance over ungrounded reasoning across both tasks and both base models. On toxicity (Table 2), tool grounding raises the LLaMA-8B F1 from 0.74 to 0.81 and the Qwen-7B F1 from 0.69 to 0.72. On mutagenicity (Table 3), the gains are similarly consistent: the LLaMA-8B model improves from 0.79 to 0.80, while the Qwen-7B model sees a more pronounced jump from 0.64 to 0.71.

---

Unless otherwise specified, all experiments , including baselines, use 50% of the training data. For Mol2Synth variants, synthetic traces are generated for this subset and filtered by the critic; only Excellent-rated traces are retained for fine-tuning.

Table 2: Performance on toxicity prediction using 50% of the synthetic training data*. Results are reported as mean $\pm$ standard deviation over three independent runs with different random seeds. While Mol2Synth stabilizes fine-tuning and improves performance over label-only supervision, even the best tool-grounded configuration remains below the structure-aware ECFP baseline. **Bold** indicates the best overall performance; underlined values denote the best fine-tuned text-only LLM-based result based on F1 score.

| Method Class | Model | Method | Accuracy | F1 |
|---|---|---|---|---|
| **Zero-shot LLMs** | | | | |
| SMILES | gemini-2.5pro | Zero-shot | $0.522 \pm 0.027$ | $0.718 \pm 0.018$ |
| SMILES | DeepSeek R1 Distill LLaMA 8B | Zero-shot | $0.610 \pm 0.079$ | $0.713 \pm 0.060$ |
| SMILES | DeepSeek R1 Distill Qwen 7B | Zero-shot | $0.578 \pm 0.030$ | $0.673 \pm 0.059$ |
| SMILES | ChemDFM-13B | Zero-shot | $0.537 \pm 0.082$ | $0.698 \pm 0.077$ |
| SMILES | ether-0-24B | Zero-shot | $0.570 \pm 0.013$ | $0.700 \pm 0.021$ |
| SMILES | GPT-4o-mini | Zero-shot | $0.722 \pm 0.081$ | $0.861 \pm 0.080$ |
| **Task-specific models (baselines)** | | | | |
| SMILES | ChemBERTa-2 | Transformer encoder | $0.776 \pm 0.012$ | $0.867 \pm 0.018$ |
| SMILES | MolFormer | Transformer encoder | $0.894 \pm 0.016$ | $0.942 \pm 0.13$ |
| SMILES | ECFP | Fingerprint | $\mathbf{0.921 \pm 0.013}$ | $\mathbf{0.965 \pm 0.022}$ |
| **SFT-Direct (label-only)** | | | | |
| SMILES | DeepSeek R1 Distill LLaMA 8B | SFT-Direct | $0.365 \pm 0.013$ | $0.478 \pm 0.025$ |
| SMILES | DeepSeek R1 Distill Qwen 7B | SFT-Direct | $0.522 \pm 0.022$ | $0.57 \pm 0.021$ |
| **SFT with synthetic CoT (Mol2Synth)** | | | | |
| SMILES | DeepSeek R1 Distill LLaMA 8B | Mol2Synth | $0.624 \pm 0.022$ | $0.736 \pm 0.018$ |
| SMILES | DeepSeek R1 Distill LLaMA 8B | Mol2Synth+tool | $0.693 \pm 0.018$ | $0.807 \pm 0.028$ |
| SMILES | DeepSeek R1 Distill Qwen 7B | Mol2Synth | $0.610 \pm 0.015$ | $0.685 \pm 0.021$ |
| SMILES | DeepSeek R1 Distill Qwen 7B | Mol2Synth+tool | $0.681 \pm 0.020$ | $0.715 \pm 0.011$ |

Table 3: Mutagenicity prediction performance (50% of the synthetic training data). SFT methods use synthetic chain-of-thought (Mol2Synth) generated by Gemini 2.5 Pro. Error bars report the standard deviation across three independent runs. **Bold** indicates the best overall performance; underlined values denote the best fine-tuned text-only LLM-based result based on F1 score.

| Method Class | Model | Method | Accuracy | F1 |
|---|---|---|---|---|
| **Zero-shot LLMs** | | | | |
| SMILES | gemini-2.5pro | Zero-shot | $0.450 \pm 0.052$ | $0.578 \pm 0.055$ |
| SMILES | DeepSeek R1 Distill LLaMA 8B* | Zero-shot | $0.638 \pm 0.081$ | $0.753 \pm 0.072$ |
| SMILES | DeepSeek R1 Distill Qwen 7B* | Zero-shot | $0.650 \pm 0.021$ | $0.710 \pm 0.010$ |
| SMILES | gpt-4o-mini | Zero-shot | $0.595 \pm 0.073$ | $0.707 \pm 0.069$ |
| SMILES | ChemDFM-13B | Zero-shot | $0.740 \pm 0.044$ | $0.613 \pm 0.051$ |
| SMILES | ether-0-24B | Zero-shot | $0.724 \pm 0.015$ | $0.690 \pm 0.001$ |
| **Task-specific models (baselines)** | | | | |
| SMILES | ChemBERTa-2 | Transformer encoder | $0.564 \pm 0.018$ | $0.708 \pm 0.019$ |
| SMILES | MolFormer | Transformer encoder | $0.557 \pm 0.017$ | $0.625 \pm 0.019$ |
| SMILES | ECFP | Fingerprint | $\mathbf{0.791 \pm 0.012}$ | $\mathbf{0.776 \pm 0.018}$ |
| **SFT-Direct (label-only)** | | | | |
| SMILES | DeepSeek R1 Distill LLaMA 8B | SFT-Direct | $0.486 \pm 0.020$ | $0.559 \pm 0.028$ |
| SMILES | DeepSeek R1 Distill Qwen 7B | SFT-Direct | $0.520 \pm 0.011$ | $0.610 \pm 0.013$ |
| **SFT with synthetic CoT (Mol2Synth)** | | | | |
| SMILES | DeepSeek R1 Distill LLaMA 8B | Mol2Synth | $0.675 \pm 0.027$ | $0.785 \pm 0.022$ |
| SMILES | DeepSeek R1 Distill LLaMA 8B | Mol2Synth+tool | $0.692 \pm 0.022$ | $0.796 \pm 0.027$ |
| SMILES | DeepSeek R1 Distill Qwen 7B | Mol2Synth | $0.623 \pm 0.011$ | $0.640 \pm 0.013$ |
| SMILES | DeepSeek R1 Distill Qwen 7B | Mol2Synth+tool | $0.752 \pm 0.021$ | $0.710 \pm 0.021$ |

**3. Impact of Representation (SMILES vs. IUPAC).** Switching from SMILES to standardized IUPAC names yields the largest single performance gain, boosting Toxicity F1 from 0.74 to 0.88 and Mutagenicity F1 from 0.79 to 0.82 (see Table 4). While this indicates that semantic nomenclature

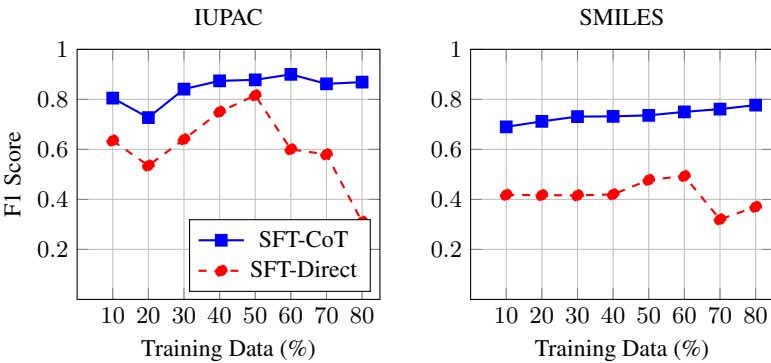

Figure 2: **Data Scaling Saturation.** IUPAC performance (left) plateaus early ($\approx 40\%$), confirming an information bottleneck where additional text supervision yields diminishing returns. In contrast, SMILES (right) suffers a persistent gap, while label-only SFT (Red) exhibits instability at scale.

better supports causal reasoning than ASCII strings, the resulting model still trails the structure-based ECFP baseline (F1 0.96), confirming that neither text format fully encodes the requisite physicochemical signal.

Table 4: Impact of different input types on downstream task performance. Unless otherwise specified, all results use DeepSeek R1 Distill LLaMA 8B as the base model. All methods apply Mol2Synth with traces generated by a LLM. Error bars report the standard deviation across three independent runs. Best F1 score is highlighted in **bold**

| Property | Input Type | Accuracy | Precision | Recall | F1 |
|---|---|---|---|---|---|
| Mutagenicity | SMILES | $0.675 \pm 0.027$ | $0.747 \pm 0.025$ | $0.872 \pm 0.021$ | $0.785 \pm 0.022$ |
| | IUPAC | $0.707 \pm 0.022$ | $0.730 \pm 0.021$ | $0.938 \pm 0.021$ | $\mathbf{0.821 \pm 0.021}$ |
| | SMILES+IUPAC | $0.701 \pm 0.022$ | $0.743 \pm 0.020$ | $0.913 \pm 0.021$ | $0.813 \pm 0.023$ |
| Toxicity | SMILES | $0.624 \pm 0.022$ | $0.926 \pm 0.021$ | $0.613 \pm 0.025$ | $0.736 \pm 0.018$ |
| | IUPAC | $0.800 \pm 0.017$ | $0.919 \pm 0.019$ | $0.847 \pm 0.010$ | $\mathbf{0.878 \pm 0.010}$ |
| | SMILES+IUPAC | $0.774 \pm 0.023$ | $0.914 \pm 0.020$ | $0.804 \pm 0.021$ | $0.856 \pm 0.026$ |

## 5.3 Qualitative Failure Analysis: Structural Blindness

To understand the mechanistic basis of performance differences across our methods, we conducted a qualitative analysis of reasoning traces produced under two conditions: **(1)** SMILES vs. IUPAC inputs for Mol2Synth, and **(2)** SMILES inputs with Mol2Synth vs. Mol2Synth+tool. Outputs were assessed for chemical correctness, reasoning consistency, and toxicological plausibility, with selected traces reviewed by a medicinal chemist (Appendix A.7). Our analysis identifies *structural hallucination* as the primary failure mode and root cause of the "SMILES gap." We make the following key observations:

**Structural misinterpretation from SMILES inputs:** When given the IUPAC name octan-3-one, Mol2Synth produced a correct and coherent analysis, identifying it as a simple aliphatic ketone with low toxicity. However, when provided the equivalent SMILES (CCCCCC(=O)CC), the **model misidentified the molecule as 3-nonanone**; an incorrect carbon count that propagated through all subsequent reasoning, yielding an internally coherent but factually wrong toxicity assessment. In contrast, the IUPAC input encodes molecular identity in a format that aligns more naturally with the language model's tokenization, reducing the opportunity for such misinterpretation. This directly explains why IUPAC inputs outperform SMILES inputs in Table 4 (Appendix A.6).

**Isomer confusion without tool-based standardization:** For CC(C)=CCl, the tool-augmented pipeline correctly standardized the structure via canonical SMILES conversion, identified a vinyl/allylic chloride toxicophore, and classified the compound as toxic. Without tools, the **model misassigned the SMILES to a structural isomer**, failing to recognize the relevant toxicophore and again applying chemically sound reasoning to the wrong molecule (Appendix A.6).

**Cascading reasoning errors from early structural faults are silent and self-consistent:** Across both case studies, early SMILES misidentification produces downstream reasoning that is chemically detailed and logically consistent; yet applied to the wrong molecule. Critically, the **model never self-corrects or signals uncertainty** about its structural assignment, making these failures particularly insidious. Medicinal chemist annotations (Appendix A.7) confirm that these **initial structural errors, rather than flaws in chemical reasoning *per se*, are the dominant source of incorrect predictions**. The reasoning quality *given* a correct structural interpretation is generally sound; it is the SMILES-to-structure mapping itself that proves unreliable.

**Implications for tool grounding and the performance ceiling:** These findings carry two important implications. First, they explain why **tool grounding yields consistent improvements** across both tasks and all base models: by offloading structural parsing to deterministic cheminformatics tools, Mol2Synth+tool eliminates the most error-prone step in the reasoning pipeline. Second, they suggest that the **performance ceiling of LLM-based molecular reasoning is not fundamentally limited by the models' chemical knowledge**, but rather by the fidelity of the initial structural interpretation; a bottleneck that tool integration can substantially alleviate but that explicit structural representations such as ECFP fingerprints bypass entirely.

## 6 LIMITATIONS

Our work has several limitations. First, our critic model evaluates traces on chemical coherence and toxicological plausibility rather than ground-truth labels, which may implicitly strengthen supervision beyond label-only fine-tuning. Second, synthetic reasoning quality is bounded by the frontier LLM used for generation (Gemini 2.5 Pro); a different teacher model may yield different performance characteristics, and evaluating Mol2Synth across a broader set of teacher models remains an important direction. Third, even with tool-grounded reasoning and IUPAC inputs, our best configuration (F1 0.88) falls short of the ECFP fingerprint baseline (F1 0.96) on toxicity, indicating that textual representations impose an information bottleneck that natural language reasoning cannot fully overcome. In future work, we plan to address this gap by grounding chain-of-thought traces with topological-derived rationales; such as ECFP bit attributions identifying which structural fragments drive predictions; combining the interpretability of natural language reasoning with the discriminative power of explicit structural features. We also note that critic-based filtering, while validated against expert annotations, introduces subjective quality thresholds that may not generalize across all chemical properties.

Finally, we evaluate Mol2Synth exclusively on binary classification tasks (toxicity and mutagenicity); its effectiveness on regression tasks such as solubility or binding affinity prediction, where continuous reasoning over quantitative structure–property relationships is required, remains unexplored.

## 7 CONCLUSION

In this work, we introduced Mol2Synth, a framework that demonstrates how synthetic chain-of-thought supervision and tool grounding can unlock meaningful molecular reasoning capabilities in small, open-weight LLMs. Our pipeline stabilizes fine-tuning and substantially improves performance, raising F1 from 0.48 under label-only supervision to 0.88 with tool-grounded reasoning and semantic inputs, while surpassing both zero-shot frontier models and the domain-specialized ChemDFM-13B, and ether-0. On mutagenicity, Mol2Synth even outperforms dedicated molecular encoders such as ChemBERTa-2 and MolFormer, demonstrating that structured reasoning traces can compete with task-specific architectures when substructure-level reasoning is central to the prediction task. Our qualitative analysis further reveals that the remaining performance gap relative to fingerprint baselines stems not from deficiencies in chemical reasoning, but from structural hallucinations during SMILES interpretations; a targeted bottleneck that tool integration already partially alleviates. These findings establish Mol2Synth as a practical and data-efficient approach for molecular property prediction, and point toward a clear path forward: hybrid architectures that combine the interpretability and flexibility of language-based reasoning with the discriminative power of explicit structural representations.

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

# A  APPENDIX

## A.1  PROMPT TEMPLATES

### A.1.1  SYNTHETIC DATA GENERATOR PROMPT (TOXICITY)

---

**Generator Prompt (Toxicity)**

**System Message:** You are a medicinal chemistry expert analyzing molecular toxicity. Given a molecule's `input_description`, predict whether it exhibits clinical toxicity. First, provide your reasoning process within `<think></think>` tags. Consider relevant factors such as structural alerts, functional groups, molecular properties, and known toxicophores. Then, choose one option and place your choice within `<answer></answer>` tags: `toxic` or `non-toxic`.

**Example Output:** `<think> [Your reasoning here] </think> <answer> toxic / non-toxic </answer>`

**User Message:** Analyze the following molecule for toxicity:
`[SMILES/IUPAC: {item["input"]}]`

---

### A.1.2  SYNTHETIC DATA GENERATOR PROMPT (MUTAGENICITY)

---

**Generator Prompt (Mutagenicity)**

**System Message:** You are a medicinal chemistry expert analyzing molecular mutagenicity. Given a molecule's SMILES or IUPAC representation, predict whether it is mutagenic (i.e., can cause genetic mutations). First, provide your reasoning process within `<think></think>` tags. Consider relevant factors such as structural alerts for mutagenicity, electrophilic groups, DNA-reactive moieties, and known mutagenic pharmacophores. Then, choose one option and place your choice within `<answer></answer>` tags: `mutagenic` or `non-mutagenic`.

**Example Output:** `<think> [Your reasoning here] </think> <answer> mutagenic / non-mutagenic </answer>`

**User Message:** Analyze the following molecule for mutagenicity:
`[SMILES/IUPAC: {item["input"]}]`

---

### A.1.3  CRITIC PROMPT (TOXICITY)

---

**Critiquer Prompt (Toxicity)**

**System Message:** You are an expert medicinal chemist and toxicologist acting as a critic. Your task is to evaluate the *reasoning process* of another AI model that was asked to predict molecular toxicity. The AI was provided with the molecule's `input_description`. **Focus ONLY on the quality of the reasoning** contained within the `<think></think>` block. *Do not judge the final answer—evaluate only the reasoning process.*

Evaluate the reasoning based on the following criteria:

- Is the reasoning chemically sound?
- Does it identify relevant structural features and toxicophores?
- Does it consider appropriate toxicity mechanisms?
- Is the logic clear and well-connected?
- Are the chemical concepts accurate?

Output your evaluation **ONLY** in the following format:

---

```
        <reasoning> [Brief justification for your
        evaluation] </reasoning>
        <evaluation> excellent / good / average / bad /
        terrible </evaluation>
```

**User Query (Critique): Original User Query:** [user query]
**AI's Reasoning (block):** [generated thinking]

**Critique Task:** Evaluate the AI's reasoning based on the criteria described above and output your evaluation in the specified format.

### A.1.4 CRITIC PROMPT (MUTAGENICITY)

---

**Critiquer Prompt (Mutagenicity)**

**System Message:** You are an expert medicinal chemist and genetic toxicologist acting as a critic. Your task is to evaluate the *reasoning process* of another AI model that was asked to predict molecular mutagenicity. The AI was provided with the molecule's `input_description`. **Focus ONLY on the quality, logical flow, and chemical/biological relevance of the reasoning** contained within the `<think></think>` block. *Do not judge the final answer—evaluate only the reasoning process.*

Evaluate the reasoning based on the following criteria:

- Is the reasoning chemically sound?
- Does it identify relevant structural features known to cause mutagenicity?
- Does it recognize DNA-reactive groups (e.g., electrophiles, alkylating agents)?
- Does it consider metabolic activation pathways?
- Are the mutagenicity mechanisms clearly and correctly explained?
- Is the logic clear and well-connected?

Output your evaluation **ONLY** in the following format:

```
        <reasoning> [Brief justification for your
        evaluation] </reasoning>
        <evaluation> excellent / good / average / bad /
        terrible </evaluation>
```

**User Query (Critique): Original User Query:** [user query]
**AI's Reasoning (block):** [generated thinking]

**Critique Task:** Evaluate the AI's reasoning using the criteria above and output your evaluation in the specified format.

---

## A.2 EXPERIMENTAL SET UP
**Datasets**

For data sets, we use two properties: mutagenicity and toxicity. The total number of data points across labels and tasks can be seen in Table 5.

Table 5: Number of datapoints across tasks

| Name | Description | Size | No. of molecules train/val/test |
|------|-------------|------|--------------------------------|
| Toxicity | Toxicity prediction | $\sim 7000$ | 4700/1353/670 |
| Mutagenicity | Mutagenicity prediction derived from the Ames test | $\sim 6000$ | 5456/504/158 |

**Baselines**

We compare our Mol2Synth approach against a diverse set of baselines spanning zero-shot large language models, task-specific molecular models, and label-only supervised fine-tuning.

- **Zero-shot LLMs** : We evaluate multiple pretrained LLM's, including Gemini-2.5 Pro, DeepSeek-R1 distilled variants, GPT-4o-mini, and chemistry aware LLM; ChemDFM-13B (25) and ether-0-24B (26), in a zero-shot setting using SMILES inputs. These baselines assess whether general-purpose language models can perform molecular toxicity and mutagenicity prediction without task-specific supervision or reasoning fine-tuning.

- **Task-specific molecular models** : We include ChemBERTa-2 (22) and MolFormer(8), representative transformer-based models trained specifically for molecular property prediction, as well as ECFP fingerprints. These baselines represent strong domain-specific approaches that do not rely on natural-language reasoning. All task specific molecular models were trained using RandomForest Classifier.

- **Label-only supervised fine-tuning (SFT-Direct)** : To isolate the effect of reasoning supervision, we fine-tune DeepSeek-R1 distilled models using task labels only, without chain-of-thought supervision. This baseline controls for performance gains attributable purely to supervised fine-tuning.

- **Reasoning-augmented supervised fine-tuning (Mol2Synth)** ;We compare label-only SFT against supervised fine-tuning with synthetic chain-of-thought rationales generated by an external LLM. We further evaluate a tool-augmented variant (Mol2Synth+tool) that incorporates structure-aware validation during rationale generation to reduce structural errors.

All baseline results are reported for toxicity in Table 2 and mutagenicity in Table 3, with error bars computed over three independent runs.

**Model Configuration**

Across all tasks, we use Gemini 2.5 Pro (29) for synthetic data generation and the DeepSeek-R1–distilled Llama 3.1 8B model for fine-tuning. Fine-tuning is performed using Low-Rank Adaptation (LoRA) (30) to reduce memory footprint and training cost while preserving model performance.

**Hyperparameters**

We trained and evaluated both SFT-Direct and Mol2Synth using Unsloth (31). Table 6 lists all custom hyperparameters used in our experiments.

Table 6: Hyperparameter settings for all experiments

| Hyperparameter | Value | Description |
|---|---|---|
| Learning rate | $1 \times 10^{-4}$ | AdamW optimizer |
| Batch size | 4 | Per GPU |
| Warmup steps | 5 | Linear warmup |
| Max sequence length | 2048 | Output truncation |
| Epochs | 50 | Training duration |

**Compute Resources**

We performed all training and evaluation on a single NVIDIA A100 (80GB) GPU using BF16 precision and the Unsloth (31) framework. Each fine-tuning run required approximately 1.5 GPU-hours.

### A.3  ADDITIONAL RESULTS

#### A.3.1  EFFECT OF TRAINING DATA FRACTION ON TOXICITY PREDICTION

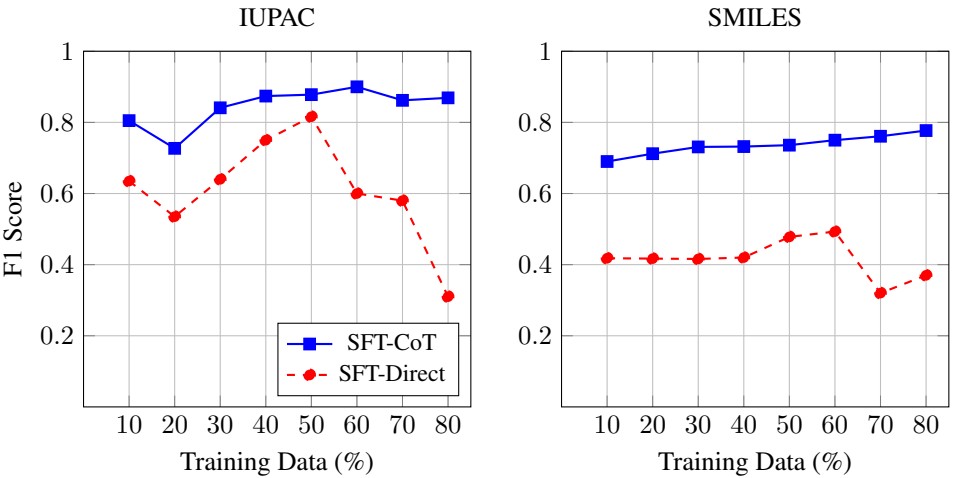

Figure 3: Effect of increasing training data on F1 score for toxicity prediction using IUPAC (left) and SMILES (right) inputs. SFT-CoT consistently improves with additional data, while SFT-Direct exhibits higher instability, particularly at larger data fractions.

Figure 3 shows the effect of increasing training data fractions (10%–80%) on F1 score for toxicity prediction using IUPAC (left) and SMILES (right) inputs. For SFT-CoT (blue line), performance saturates relatively early: IUPAC reaches high F1 by 30–40% (0.841–0.874) and then largely plateaus (0.878 at 50%, peak 0.900 at 60%), while SMILES shows only modest gains beyond 20–30% (0.712–0.731) and remains in a narrow band thereafter (0.732–0.777).

In contrast, the label-only SFT-Direct baseline (red dashed line) is substantially less stable: for IUPAC it peaks at 0.816 (50%) and then drops sharply to 0.310 (80%), and for SMILES it remains low, with a noticeable degradation at higher data fractions.

These results suggest that synthetic CoT supervision improves performance and robustness relative to label-only fine-tuning, especially at higher data fractions where SFT-Direct can become unstable. We hypothesize that this instability reflects sensitivity of label-only fine-tuning to class imbalance and decision-thresholding (which strongly affects F1), as well as optimization/seed sensitivity when supervision is limited to a single label.

#### A.3.2  RESULTS:CRITIC FILTERING THRESHOLD ANALYSIS

Table 7: Effect of critic filtering threshold on Mol2Synth toxicity prediction performance using SMILES inputs (50% training data).

| Minimum rationale quality | Accepted critic tags | F1 |
|---|---|---|
| 5/5 (strict) | excellent | **0.73** |
| 4/5 | excellent, good | 0.71 |
| 3/5 | excellent, good, average | 0.71 |
| 2/5 | excellent, good, average, bad | 0.67 |
| 0–1/5 (no filtering) | excellent, good, average, bad, poor | 0.64 |

### A.4  CHEMISTRY INFORMED REASONING TOOLS

#### A.4.1  SMILES CLEANING

This tool cleans up SMILES strings by removing those that cannot be accurately converted into a valid molecular format using rdkit. Eventually, kekulization was applied to the valid SMILES,

which involved converting aromatic bonds into alternating single and double bonds, thereby creating a more standardized representation of the molecular structure.

### A.4.2 TOXICOPHORE DETECTOR

The Toxicophore Detector identifies known toxicophoric substructures within the provided SMILES string. We then, match these substructures to a database of common toxicophores from the OChem-Database, facilitating the assessment of potential toxicity associated with the molecular structure.

### A.4.3 MUTAGENIC DETECTOR

Similar to the toxicophore detector, the Mutagenic Detector identifies known mutagenic substructures within the provided SMILES string. We then, match these substructures to a database of common mutagens from the OChemDatabase, facilitating the assessment of potential mutagens associated with the molecular structure.

### A.4.4 SCAFFOLD SEARCH

The Scaffold Search tool analyzes the provided SMILES string to identify relevant scaffolds linked to known chemical toxicophore or mutagenic entities in the literature. By recognizing and extracting these scaffolds, researchers can better understand the structural frameworks that underlie biological activity and potentially discover novel compounds through scaffold hopping and optimization.

### A.5 GLOSSARY OF ABBREVIATIONS

| Abbreviation | Full Form |
|---|---|
| SFT | Supervised Fine-Tuning |
| CoT | Chain-of-Thought |
| LLM | Large Language Model |
| LoRA | Low-Rank Adaptation |
| SMILES | Simplified Molecular Input Line Entry System |
| IUPAC | International Union of Pure and Applied Chemistry |
| SFT-CoT | Supervised Fine-Tuning with Chain-of-Thought |
| CI | Confidence Interval |
| Temp | Temperature |
| ECFP | Extended Connectivity Fingerprints |

Table 8: List of abbreviations used in the paper.

A.6    FAILURE DIAGNOSIS OF REASONING TRACES

## Case Study 1: Input Representation Failure (IUPAC vs SMILES)

---

**Problem / User Query**

Given a molecular input, can you determine whether the molecule is toxic or non-toxic?

---

**IUPAC Input**

**Input:** Octan-3-one

**SMILES Input**

**Input:** CCCCCCC(=O)CC

---

| Observation | Result |
|---|---|
| Molecule identified | Octan-3-one |
| Structural correctness | ✓ |
| Reasoning consistency | High |
| Toxicity conclusion | Low risk |

| Observation | Result |
|---|---|
| Molecule identified | 3-nonanone (incorrect) |
| Structural correctness | ✗ |
| Reasoning consistency | Cascading errors |
| Toxicity conclusion | Unreliable |

---

**Summarized CoT (IUPAC)**

Octan-3-one is correctly identified as a simple aliphatic ketone with an eight-carbon chain. It lacks known toxicophores, including $\alpha, \beta$-unsaturated carbonyls and neurotoxic $\gamma$-diketones. Expected metabolism involves ketone reduction and $\omega/\omega$-1 oxidation, producing more polar, readily excreted metabolites. Physicochemical properties and established GRAS usage support a low toxicity profile.

**Summarized CoT (SMILES)**

The SMILES string is misinterpreted as a different ketone (3-nonanone), leading the model to analyze an incorrect molecular structure. Reasoning focuses on general ketone metabolism and the absence of known neurotoxic $\gamma$-diketone formation. Although chemically coherent, the reasoning applies to the wrong molecule, resulting in an internally consistent but unreliable toxicity assessment.

---

**Key Observation**

**IUPAC-based inputs preserve correct molecular identity and lead to consistent reasoning.** In contrast, raw SMILES inputs can cause molecule misidentification, resulting in cascading factual errors despite chemically plausible reasoning.

## Case Study 2: Tool Prompting vs No Tool Prompting

**Problem / User Query**

Given the SMILES `CC(C)=CCl`, can you determine whether the molecule is toxic or non-toxic?

**With Mol2Synth+tool**

**Step 1:** Cleaned SMILES: `CC(C)=CCl`
**Step 2:** Toxicophore analysis Vinyl chloride–like fragment identified.
**Step 3:** Fragment analysis Allylic chloride moiety detected.
**Conclusion:** Classified as toxic.

**Without Tool Prompting**

Incorrect molecule identification due to SMILES isomer confusion, leading to erroneous functional group assignment and cascading reasoning errors.

| Observation | With Tool |
|---|---|
| SMILES cleaned | ✓ |
| Molecule identified | Correct |
| Fragment mapping | Correct |
| Reasoning quality | Consistent |
| Final classification | Toxic |

| Observation | Without Tool |
|---|---|
| SMILES cleaned | × |
| Molecule identified | Incorrect |
| Fragment mapping | Incorrect |
| Reasoning quality | Cascading errors |
| Final classification | Unreliable |

**Summarized CoT (With Tool)**

The molecule is correctly identified as methallyl chloride (3-chloro-2-methylprop-1-ene), containing an allylic chloride toxicophore. This functional group is a known electrophile capable of alkylating biological nucleophiles. Reasoning focuses on covalent DNA and protein interactions, glutathione depletion, and known toxicity of related allylic chlorides, leading to a consistent classification as toxic.

**Summarized CoT (Without Tool)**

The SMILES is misinterpreted as a vinyl halide, triggering analysis based on an incorrect molecular structure. Reasoning emphasizes epoxide formation and analogy to vinyl chloride carcinogenicity. Although chemically coherent, the reasoning applies to the wrong toxicophore, resulting in internally consistent but structurally misaligned toxicity conclusions.

**Key Observation**

Tool prompting enables SMILES cleaning and accurate molecular identification, anchoring reasoning to the correct toxicophore and stabilizing downstream inference. Without tool prompting, early misidentification propagates through the reasoning chain, leading to chemically plausible but structurally incorrect toxicity assessments.

## A.7 EXPERT ANNOTATION OF REASONING TRACES

We ask a medicinal chemist to annotate the reasoning traces. Below, we show examples for toxicity prediction using Mol2Synth (Figure 4) and Mol2Synth+tool (Figure 5). We observe that the medicinal chemist annotations (Figure 4) highlight how factual errors propagate through the reasoning chain once the initial statement is incorrect. When Mol2Synth+tool is introduced, the SMILES are first cleaned, enabling the LLM to correctly identify the structure of the SMILES respective and thus identify the molecule.

---

**Prompt**

Analyze the following molecule for clinical toxicity given the following SMILES CCCC1=CC=C(OC)C=C1

---

**Expert Annotation**

- **The molecule is 1-butyl-4-methoxybenzene**, characterized by a benzene ring substituted with a butyl group and a methoxy group. **(FALSE)**

- Structural Alerts/Toxicophores: Benzene ring: While benzene itself is a known carcinogen, its toxicity is significantly altered by substitution. In this case, it's disubstituted, which generally reduces its intrinsic toxicity, especially when substituents are not highly activating towards harmful oxidation (e.g., anilines). (TRUE)

- Comparison to Known Toxicophores/Analogues:The molecule lacks highly reactive functional groups such as epoxides, quinones, Michael acceptors, or highly activated aryl amines that are often associated with toxicity. (TRUE)

- The molecule is relatively simple and lacks features commonly associated with high acute or chronic toxicity. Hence (non-toxic) (TRUE)

Figure 4: Prompt and expert annotation of Mol2Synth for toxicity prediction with SMILES as input

---

**Prompt**

Analyze the following molecule for clinical toxicity given the following SMILES CCCC1=CC=C(OC)C=C1

---

**Expert Annotation**

- **Cleaned SMILES Canonical SMILES**: CCCc1ccc(OC)cc1 **Structure**: Para-substituted anisole with an n-propyl side chain Name: 1-propyl-4-methoxybenzene **(TRUE)**

- Toxicophore Analysis: No toxicophores detects; no electrophile or reactive functional groups.(TRUE)

- Fragment Analysis: Benzene ring : Hydrophobic scaffold; propyl chain; chemically inert (TRUE)

- The molecule is relatively simple and lacks features commonly associated with high acute or chronic toxicity. Hence (non-toxic) (TRUE)

Figure 5: Prompt and expert annotation of Mol2Synth+toolfor toxicity prediction with SMILES as input

