# OpenReview forum: "Understanding The Limits Of Text-Only Molecular Reasoning: A Case Study In Synthetic Chain-Of-Thought Supervision"
_ICLR.cc/2026/Workshop/FM4Science — ICLR 2026 Workshop FM4Science Poster_

### Official Review · Reviewer_zcmB · 2026-02-19

**Rating:** 7
**Confidence:** 4

**Review:**

### Evaluation

**Originality:**
The paper frames synthetic CoT supervision as a lightweight alternative to reinforcement learning for scientific reasoning tasks. The focus on identifying and diagnosing the limits of text-only reasoning is a valuable angle.

**Quality:**
The experimental design is comprehensive. Multiple representations, two base models, two supervision types, and two generation settings are compared.

**Clarity:**
The paper is clearly written and well-structured.

**Significance:**
The paper makes both positive and negative contributions that are likely to influence future work. The paper thus serves as both a practical guide for practitioners and a conceptual contribution for researchers.


### Pros

1. The study compares multiple representations, models, supervision types, and baselines, with statistical replicates and ablations. This makes the findings robust and trustworthy.
2. The paper clearly documents the performance gap relative to ECFP on toxicity and traces this gap to structural hallucination.
3. The qualitative failure analysis identifies the specific bottleneck and shows how tool grounding alleviates it.
4. The paper is well-organized, with a logical flow from motivation to methods to results to limitations. The figures and tables are informative.

### Cons

1. The synthetic traces are generated only by Gemini 2.5 Pro. The paper would be strengthened by ablating across multiple teacher models (e.g., GPT-4, Claude) to assess Mol2Synth's sensitivity to the generator's capabilities.
2. The study focuses exclusively on binary classification (toxicity, mutagenicity). It is unclear how well Mol2Synth would generalize to regression tasks (e.g., solubility, binding affinity) that require continuous reasoning about quantitative structure-property relationships.
3.  The critic uses a five-criterion rubric rated by an LLM, which introduces a subjective quality threshold. While the authors validate this against expert annotations, the threshold remains somewhat arbitrary and may not generalize to all chemical properties.

---

### Official Review · Reviewer_A6zA · 2026-02-22
**A controlled study (Mol2Synth) that demonstrates both the promise and fundamental limits of synthetic chain-of-thought supervision for text-only molecular reasoning, with the insight that the performance ceiling stems from SMILES-to-structure parsing failures rather than deficient chemical reasoning.**

**Rating:** 7
**Confidence:** 2

**Review:**

This paper introduces Mol2Synth, a framework for augmenting LLM fine-tuning with synthetic CoT reasoning traces for toxicity/mutagenicity prediction, comparing SMILES vs. IUPAC representations with and without tool grounding.

Pros:

(1) Well-controlled experimental design with systematic ablations across representations, data scale, and tool grounding.

(2) Key insight is valuable: the performance gap vs. ECFP fingerprints (F1 0.88 vs. 0.96) stems from structural parsing hallucinations, not faulty chemical reasoning.

(3) Excellent qualitative failure analysis with expert chemist annotations.

(4) Practical—runs on a single A100, surpasses domain-specialized 13B/24B models.

(5) Honest about limitations.

Cons:

(1) Only two binary classification tasks (toxicity, mutagenicity); generalizability unclear.

(2) Synthetic traces generated by Gemini 2.5 Pro—teacher model dependency unexplored.

(3) The "negative result" (can't match ECFP) is somewhat expected given text lacks explicit topology.

(4) Limited molecular diversity analysis.

---

### Decision · Program_Chairs · 2026-03-03

Accept (Poster)